# Polyploidization: A Biological Force That Enhances Stress Resistance

**DOI:** 10.3390/ijms25041957

**Published:** 2024-02-06

**Authors:** Xiaoying Li, Luyue Zhang, Xiaochun Wei, Tanusree Datta, Fang Wei, Zhengqing Xie

**Affiliations:** 1Henan International Joint Laboratory of Crop Gene Resources and Improvements, School of Agricultural Sciences, Zhengzhou University, Zhengzhou 450001, China; 2Institute of Horticulture, Henan Academy of Agricultural Sciences, Graduate T & R Base of Zhengzhou University, Zhengzhou 450002, China

**Keywords:** polyploidy, stress resistance, abiotic stress, biotic stress

## Abstract

Organisms with three or more complete sets of chromosomes are designated as polyploids. Polyploidy serves as a crucial pathway in biological evolution and enriches species diversity, which is demonstrated to have significant advantages in coping with both biotic stressors (such as diseases and pests) and abiotic stressors (like extreme temperatures, drought, and salinity), particularly in the context of ongoing global climate deterioration, increased agrochemical use, and industrialization. Polyploid cultivars have been developed to achieve higher yields and improved product quality. Numerous studies have shown that polyploids exhibit substantial enhancements in cell size and structure, physiological and biochemical traits, gene expression, and epigenetic modifications compared to their diploid counterparts. However, some research also suggested that increased stress tolerance might not always be associated with polyploidy. Therefore, a more comprehensive and detailed investigation is essential to complete the underlying stress tolerance mechanisms of polyploids. Thus, this review summarizes the mechanism of polyploid formation, the polyploid biochemical tolerance mechanism of abiotic and biotic stressors, and molecular regulatory networks that confer polyploidy stress tolerance, which can shed light on the theoretical foundation for future research.

## 1. Introduction

Polyploid, defined as the possession of three or more sets of chromosomes resulting from whole-genome duplication (WGD) [1], plays a significant role in both evolutionary innovation and species diversification throughout evolutionary history [2,3]. Polyploidy is widespread in nature and has been previously regarded as an evolutionary ‘dead-end’ [4] or an occasional incidence [5]; however, the latest perspective on polyploidy’s evolutionary fate has been shifted towards viewing it as a driver of contemporary innovation [2,6]. Autopolyploidy arises from the duplication of genomes within a single species, while allopolyploidy involves the hybridization and genome merging between different species [7]. Generally, autotetraploids may exhibit greater resistance than diploids due to the higher expression of defense-related genes, a consequence of the doubled genome [8]. Similarly, allopolyploids allow for the utilization of additional dominant wild-type alleles to mask the deleterious ones. Additionally, heterosis or hybrid vigor can be observed in allopolyploids with gene expression alteration resulting from a combination of genome dosage regulation, epigenetic modification, and allele interaction [9]. Among plants, polyploidy is highly prevalent, with approximately 70% of angiosperms being polyploid, including autopolyploid species like *Medicago sativa* (alfalfa) and potato, as well as allopolyploid species such as wheat, cotton, rapeseed, oats, and coffee [10,11]. Almost all flowering plants have experienced at least one WGD event [12], with ancient polyploid species like maize, rice, and cabbage as notable examples [13]. However, the establishment or long-term survival of many WGDs is not random, and it often coincides with the dominating periods of global climatic/geologic change or mass extinction [2]. Although these ancient WGDs occurred independently, they have likely contributed to the expansion of many stress-related genes, such as heat shock transcription factors and Arabidopsis response regulators, which could have been selected for global environmental changes in the Cretaceous Period [14]. Interestingly, the nucleotypic effect of genome size and the extra existence of genomic materials in polyploids confer high genomic plasticity and complexity, providing them significant evolutionary advantages over their diploid counterparts. Furthermore, as a driver of reproductive isolation, phenotypic diversity, and speciation, polyploidy has an unparalleled role in generating plant diversity [15].

WGD or polyploidization has been considered a primary driver in the evolution of genes, including the emergence of genes with new functions [16,17]. Global or local genome duplication produces redundant copies, and some of these accumulate mutations, freeing them from the existing evolutionary and functional constraints, leading to gene function diversification [18]. The paralogous copies may acquire new functions (neo-functionalization), distribute the functions controlled by the ancestral gene between the duplicated genes (sub-functionalization), or a redundant copy may accumulate mutations and lose its function, becoming a pseudogene (non-functionality) [19]. An increase in the number of polyploids, coupled with higher speciation rates and species richness in certain groups, such as the ~400 species of *Draba* within the Brassicaceae, was influenced by the Late Pliocene and alternating cold and warm phases during Pleistocene glaciation and deglaciation cycles. This suggested that polyploidization in Brassicaceae was a prerequisite for subsequent diversification and radiation [20]. While polyploidy is prevalent, not all polyploidizations can successfully lead to species diversification over the long-term evolution [21,22]. Polyploid plants exhibit distinct advantages when confronted with climate change and extreme conditions, primarily due to the stable maintenance of reproductive output under heightened abiotic stress and fixed differences in seed development [15]. Polyploidy profoundly impacts plant breeding, including the augmentation of organ size (the ‘gigas’ effect), mitigation of deleterious mutations, increased heterozygosity, enhancement of hybrid vigor and product quality, heightened tolerance to biotic and abiotic stressors, facilitation of gene transfer, reduction in fertility, and production of seedless varieties [23]. Moreover, polyploidization offers numerous potential adaptive benefits at the cellular level, such as increased cell size and biomass without compromising cellular and tissue structure and enhanced cellular longevity through improved tolerance to genomic stress and apoptotic signals [24]. Studies on Brassica crops have demonstrated that heterologous polyploid cultivars exhibit superior tolerance to abiotic stressors compared to diploid cultivars [25]. In *Populus ussuriensis*, physiological and biochemical indices associated with drought resistance are more favorable in polyploids than in diploids, resulting in an increased drought tolerance [26]. In addition, the whole genome triplication (WGT) also generated more copies of stress resistance genes: in *Hibiscus hamabo*, over 95% of the significantly expanded gene families belonged to recent WGT-derived genes, including multiple stress resistance-related genes, such as those in the SOS pathway and ethylene-related genes [27]. Therefore, from an agricultural standpoint, duplicated genes resulting from polyploidy have appeared as a key to crop domestication and the evolution of stress resistance [28].

Emerging data from various life forms suggest profound shifts in numerous cellular processes following polyploidization. Cellular and organismal polyploidy may manifest as a consequence of environmental stress (i.e., increased rates of polyploidization) and also as an adaptation to it. This close association with stress underscores the growing importance of studying polyploidy as cells, tissues, and whole organisms of polyploids can respond to rapid changes in their biotic and abiotic environments. Thus, we review the relationship between polyploidy (autopolyploidy and allopolyploidy) and stress, focusing on the formation and survival of polyploids in this work. Then, we highlight recent studies that enhance our understanding of how polyploids might better adapt to changing environments, why their responses to biotic agents differ from those of diploids, and how they acquire tissue-specific or temporally segregated functions (sub-functionalization). Additionally, we analyze the mechanisms underlying polyploid resistance from cellular, physiological, biochemical, gene expression, and epigenetic modification perspectives. This review will serve as a reference for further research into polyploid stress resistance and offer a theoretical foundation for polyploid breeding and its practical applications.

## 2. Stress-Induced Polyploidy

Polyploidy can be induced by natural WGD, abiotic stressors, and even artificial agents. Generally, WGD occurs through somatic chromosome doubling (somatic polyploidization) or sexually via gametic nonreduction (sexual polyploidization). In somatic polyploidization, the ectopic formation of polyploid tissue occurs through endomitosis, nuclear fusion, or c-mitosis, eventually leading to the stable establishment of polyploidy. Conversely, sexual polyploidization involves the generation of polyploids by the formation and fusion of diploid gametes, namely pollen or eggs carrying the somatic chromosome number instead of the gametophytic number, which are also referred to as diplogametes or 2n (2n = 2x) gametes [29]. In addition, environmental stresses, such as extreme temperatures, drought, or salt stress, can also induce meiotic disturbances at the biological level, leading to the production of unreduced gametes and subsequent biological genome doubling, ultimately resulting in the generation of polyploid strains [30,31,32,33]. For a detailed route description of the polyploid formation mechanism at the cytological level, please refer to Ramsey and Schemske’s work [33].

Polyploids may possess certain advantages under stress conditions, as their slower growth rate allows them to reduce resource utilization and minimize cellular damage [34]. For instance, in Rangpur lime, tetraploid rootstock exhibits elevated levels of abscisic acid (ABA) and constitutive expression of drought-responsive genes compared to diploid rootstock. Moreover, tetraploid roots display enhanced drought tolerance even when grafted with diploid scion [35]. Hexaploid *Cochlearia danica* L., an invasive species rapidly spreading in the UK and mainland Europe, has thrived particularly well on salt-alkalized roads since the 1970s and survived under the highest salt concentration conditions in the road gravel. Notably, tetraploid *C. danica* are found in saline-alkali soil, while hexaploids are prevalent in higher saline-alkali environments. However, it should be noted that tetraploids generally exhibit lower salt tolerance than diploids [36]. Additionally, studies have indicated that during the Cretaceous–Paleogene (K-Pg) boundary, a period marked by global cooling and darkness, gene duplication events are preserved in many lineages, resulting in ancient polyploidy [37]. In certain species, artificial induction of chromosome doubling is significant in their breeding process, particularly when spontaneous doubling occurs at a relatively low frequency. Moreover, chemical agents such as colchicine, oryzalin, amiprophos-methyl, pronamide, and trifluralin have been used for chromosome doubling studies in various species, with colchicine being the oldest and most widely used anti-mitotic agent [38]. Therefore, all these polyploid induction pathways are generated by abiotic environmental stresses (abiotic elicitor), in other words, stress (Figure 1). Hence, stress-induced polyploidization can be viewed as an advanced genetic adaptation mechanism, where an increase in chromosomal number within cells facilitates the enrichment of genetic material, enhancing an organism’s resistance and adaptability to ecological pressures. This process is crucial in elucidating the molecular mechanisms underlying species’ responses to environmental changes, maintaining genetic diversity, and developing crop varieties with high adaptability and productivity in agricultural biotechnology.

## 3. Polyploidy Resistance and Stress

Multiple mechanisms have been identified to elucidate the resistance of polyploidy to various adversities. These stresses can be categorized into two distinct classes based on their developmental specificity: abiotic and biotic stresses. Abiotic stresses encompass a range of conditions such as low temperature, high temperature, drought and salt stress, and even metal ions. On the other hand, biotic stress primarily pertains to the challenges posed by pests and diseases.

### 3.1. Cellular, Physiological, and Biochemical Tolerance in Response to Stress

#### 3.1.1. Polyploidy and Abiotic Stress


Cold stress


The majority of studies indicate that polyploid plants exhibit enhanced survival capabilities in cold environments. Different cytotypes occupy distinct niches; for example, the mean annual temperature is lower for octoploid than for hexaploid *Saxifraga rosacea*, leading to complete spatial segregation of the two cytotypes [39]. Additionally, studies showed that the formation of ice lenses within lacunas between the palisade tissues and spongy parenchyma tissues in the leaves of alpine tetraploid populations in mountain ranges might be one reason for their higher freezing resistance in comparison with diploids [40]. Furthermore, by flowering later, tetraploid *Ranunculus kuepferi* allocates more nutrients to carpel and ovule production and is better suited to low temperatures [41]. In addition, stomatal closure induced by low temperatures resulted in decreased net photosynthesis (P_n_). However, faster and more significant declines in P_n_, stomatal conductance (*g*_s_), chlorophyll fluorescence (F_v_/F_m_), and starch levels were observed in diploids compared to tetraploid Carrizo citrange rootstocks, along with lower levels of malondialdehyde and electrolyte leakage in tetraploids. In addition, activities of catalase (CAT), ascorbate peroxidase (APX), and dehydroascorbate reductase (DHAR) were also higher in tetraploids during cold stress, possibly due to the enhanced intake of ABA and cytokine from root to shoot [42]. In addition, cold tubers from autotetraploid potato lines had lower levels of reducing sugars compared to those from diploid lines, and a lesser increase in reducing sugars after cold treatment [43]. Moreover, elevated carbohydrate abundances during cold stress, including sucrose, glucose, fructose, hexose phosphates, fructan, raffinose, and arabinose, led to the varying intensities of oxidative stress in bread (Norstar) wheat plants compared to durum (SRN and Gerdish) wheat plants, suggesting a metabolic/regulatory capacity with a decrease in the electrolyte leakage index (ELI) and increased defense activities [44]. Furthermore, the more abundant the hexose phosphates in Norstar than in durum wheat, like Gerdish, under cold stress, the higher the reactive oxygen species (ROS) scavenging capacity, indicating an increased cold tolerance in bread wheat plants (Norstar) [44]. Moreover, the polyploidization of Escallonia did not harm the cold tolerance but significantly increased the cold tolerance [45].


2.Heat stress


Furthermore, polyploid plants are demonstrated to have enhanced resistance to high temperatures. Elevated temperatures not only effectively induce polyploid formation [46] but also sometimes affect the deposition of the outer wall of pollen [47], thereby negatively impacting the stability of meiotic recombination and heat tolerance [48]. High temperatures directly influence the growth of reproductive organs, leading to yield reduction; however, *Brassica napus* L. exhibited higher dry matter (DM) accumulation and seed yield compared to *Brassica rapa* L. and *Brassica juncea* L. under high-temperature stress [49]. Additionally, parental heat stress treatment in durum wheat positively impacted progeny traits, including chlorophyll content, grain weight and total starch content, partly by reducing damage to the photosynthetic apparatus in the offspring [50]. In addition, in quaking aspen (*Populus tremuloides* Michx.), triploids had larger leaves with more chlorophyll, higher leaf mass per area (LMA) and corresponding increases in leaf %N, nitrogen content (mmols m^−2^), ribulose biophosphate (RuBP, or *J*_max_), photosystem II (Φ_PSII_), *g*_s_ and intrinsic water-use efficiency (iWUE) compared to diploids, suggesting that structural differences due to ploidy alteration affect plant function in response to environmental stress [51]. In addition, both the quantum yield of net CO_2_ fixation (*Φ*) and the rate of light-saturated net photosynthesis (*A*_NS_) were higher in triploid aspen than in the diploid one [52]. For the perennial grass, *Themeda triandra* Forssk in Australia, tetraploid plants with heavier seeds and longer hygroscopic awns showed increased fitness in extreme environments. Thus, it is possible that the combination of elevated genome size and reproductive elasticity contributes to the benefits of polyploidy in plants during stressful time [15]. Collectively, these studies provide compelling evidence that polyploid plants exhibit heightened thermotolerance.


3.Drought stress


Polyploid organisms often exhibit raised survival rates with expanded distribution ranges, and enhanced drought resistance in dry ecological environments. For instance, after 21 days of drought stress, tetraploids in *Citrus wilsonii* showed less water loss and smaller cellular damage compared to diploids. These tetraploid plants had greener and larger leaves due to the higher chlorophyll content and larger cells, which were instrumental in retaining leaf water and reducing transpiration loss. Additionally, under drought stress, tetraploids exhibited a smaller decrease in photosynthetic parameters, such as P_n_, *g*_s_, intercellular CO_2_ concentration (C_i_), F_v_/F_m_, and chlorophyll content, than diploids, indicating that tetraploids maintain a higher photosynthetic activity under water deficit condition [53]. Thus, polyploids have larger fresh-to-dry weight ratios compared to diploids, and the higher water content indicates a more excellent osmotic adjustment capacity or a higher elasticity of cell walls, suggesting that polyploids might have a greater regulatory capability over the osmotic potential under drought stress [54]. In addition, polyploidization enhances the CO_2_ assimilation rate per leaf area and boosts the expression and activity of RuBisCO and other enzymes involved in the Calvin Cycle, thereby augmenting carbohydrate production. The var. *maritima* tetraploids produced more biomass by efficiently utilizing carbohydrates, contributing to faster plant growth than diploids [55].

ROS are toxic molecules produced in cells, while superoxide dismutase (SOD) and peroxidase (POD) are antioxidant enzymes that play a vital role in eliminating O_2_^•−^ and H_2_O_2_, respectively. Tetraploids in *Poncirus trifoliata* had significantly higher POD and SOD activities compared to diploids, responding to lower accumulation of O_2_^•−^ and H_2_O_2_ in tetraploids, which suggests the increased drought stress tolerance in autotetraploids [56]. Furthermore, tetraploids also exhibited higher cell membrane permeability (CMP, a biomarker of oxidative stress) than diploids, likely due to the fact that high levels of glutathione, ascorbate, and antioxidant enzymes can balance oxidative conditions [55]. In addition, DHAR, which regenerates ascorbic acid (AsA) via the Halliwell–Asada pathway and adjusts the cellular ascorbate redox state, had higher activity in tetraploids than diploids, contributing to an enhanced antioxidant capacity [57]. Furthermore, levels of ABA and ROS were higher in Arabidopsis autotetraploids than in diploids, potentially leading to more closed stomata, thereby contributing to decreased transpiration rates and ultimately enhancing tetraploids’ tolerance to drought stress [58]. Additionally, ABA can significantly promote the production of osmotic proteins, therefore increasing stress tolerance. In *Lycium ruthenicum*, autotetraploids showed a 78.4% increase in ABA content under natural conditions compared to diploids, suggesting a better stress tolerance in autotetraploids [59]. Invertase (INV) in plants irreversibly hydrolyzes sucrose into glucose and fructose. Thus, the elevated levels of INV in tetraploids led to reduced sucrose content and increased glucose content, suggesting that the higher glucose accumulation in autotetraploids might enhance their drought resistance [56]. Moreover, proline can effectively scavenge singlet oxygen and hydroxyl radicals, preventing lipid peroxidation under stress conditions. Therefore, the higher proline accumulation in triploid Citrus, along with lower oxidative marker accumulation, indicates a better regulatory activity of cell redox status for the protection of thylakoid membrane integrity during water deficits [57].

In conclusion, photosynthetic activity, antioxidant ability, and resistance to osmotic stress were all regulated in polyploids, thus contributing to their drought tolerance.


4.Salt stress


Soil salinity is an important abiotic stress that limits plant growth and development, negatively influencing morphology, photosynthetic efficiency, chloroplast structure, and mitochondria function [60]. Chlorophyll, the basis of photosynthesis and an index of leaf senescence, often showed with total Chl and Chl a, as well as the Chl a/Chl b ratio, were lower in diploid black locust under salt stress than in controls. In contrast, the contents of total Chl, Chl a, and Chl b were increased in tetraploids under salt stress, suggesting that tetraploid plants had a stronger ability to adapt to salt stress [61]. In addition, during salt stress, leaf photosynthetic rate and carboxylation efficiency were significantly decreased, but the decrease was less in tetraploid honeysuckle than in diploids. In addition, photosystem II (PSII) and photosystem I (PSI) photoinhibition were not found in tetraploids, possibly because the increased non-photochemical quenching (NPQ) helped dissipate excess excitation energy and prevented damage from ROS formation. By comparison, PSII and PSI underwent photoinhibition under salt stress in autodiploid honeysuckle, with significantly decreased F_v_/F_m_ and maximal photochemical capacity of PSI (MR/MR_0_) [62]. Additionally, the accumulation of Na^+^ negatively affects RuBisCO, leading to CO_2_ fixation inhibition and an increase in ROS generation, which inactivates PSII and PSI. In tetraploid *Plumbago auriculata* Lam., the rates of Na^+^ and Cl^−^ secretion were higher after salt stress than in diploids, suggesting a better salt-secreting ability in tetraploids under salt stress. Besides, tetraploid recretohalophyte necessitates the capability to sequester Na^+^ and Cl^−^, either through accumulation in leaf cell vacuoles or unloading by roots, and selectively excretes these ions via salt glands, together with the ability to prevent K^+^ loss from the roots. This regulates ion homeostasis in plants, maintaining K^+^/Na^+^ homeostasis, facilitating the salt stress tolerance in polyploids, and providing new insights into promoting ion homeostasis maintenance in polyploid plants [63].

Elevated accumulations of H_2_O_2_ and malondialdehyde (MDA) were also observed under salt stress (Figure 2). The greater H_2_O_2_ accumulation is often followed by a significant increase in essential antioxidative enzymes and non-enzymatic antioxidants [60]. In addition, SOD and POD are important enzymes against oxidative stress, which were also generally detected with higher activity levels in polyploids under salt stress. Higher H_2_O_2_ content was found in chloroplasts of 4x (2n = 4x) leaves than that of 2x leaves in black locust (*Robinia pseudoacacia* L.) in a normal environment [61]; however, the H_2_O_2_ levels were unchanged with increased salinity stress [64]. Conversely, H_2_O_2_ levels in roots of 6x *Ipomoea trifida* were considerably lower than that in 2x plants [65], and H_2_O_2_ and MDA contents were also significantly lower in tetraploids than in diploid *Plumbago auriculata* Lam. [63]. However, the contents of SOD and POD were higher in triploid than that in tetraploid and diploid *Populus ussuriensis* Kom, indicating a possibly higher salinity tolerance in triploids than tetraploid and diploid ones, suggesting that heterosis might help in triploid salt stress [66]. Additionally, tetraploid leaves had higher levels of SOD activity during salt stress, mainly relying on the Halliwell–Asada pathway in response, with greater activities of AsA, glutathione (GSH), grana (GR), and glutathione S-transferase (GST) in tetraploids [67]. APX, DHAR, and MDHAR activities were increased with H_2_O_2_ accumulation in tetraploids [67], while SOD and APX activities were decreased in diploids [64]. In addition, cytochrome c oxidase (COX) mainly participates in the pathway for ATP production from ADP and inorganic P (Pi) through oxidative phosphorylation, and the COX activity in 4x mitochondria was higher on a small scale than in 2x, indicating a higher salt tolerance in 4x mitochondria [64]. Moreover, alcohol dehydrogenase can detoxify highly reactive and toxic molecules, and its accumulation induced by salt stress could inhibit plant growth, but the levels in 4x plants were lower under salt stress, suggesting that a decline in this protein could promote tetraploid plant growth and development under salt stress [67].

Additionally, proline was reported to be involved in osmotic regulation, ROS purge, and defense of proteins against denaturation. Hence, higher proline levels detected in roots and leaves of tetraploid genotypes [68] might demonstrate a greater ability to resist the osmotic pressure induced by salt stress, indicating that 4x genotypes might have greater tolerance to salt stress. Heat shock proteins are an important factor that protects plants against stress-induced damage. However, a heat shock protein (spot 47) was down-regulated in both diploids and tetraploids [67], while the heat shock protein 70 (HSP70) was clearly induced in 4x mitochondria [64], suggesting that tetraploid plants might have enhanced tolerance to salt stress with the chaperone protein.


5.Heavy metal ions stress


Heavy metal ions are essential micronutrients at low concentrations but predominantly have toxic impacts on plant growth at elevated levels, thus disrupting normal physiological and biochemical plant processes. For example, cuprum (Cu, copper), as a cofactor in numerous physiological processes, is essential for normal plant metabolism, yet higher accumulation of this micronutrient can be extremely toxic to plants. When exposed to excess Cu, tetraploid Arabidopsis plants showed lower accumulations of H_2_O_2_, O_2_^•−^, and MDA in response to stress compared to diploids. Furthermore, the contents of GSH and the activities of antioxidant enzymes were relatively higher in autotetraploids than in diploids, implying that autotetraploids possess a stronger capacity for antioxidant responses under heavy metal ion stress [69]. Additionally, excess manganese (Mn) can inhibit plant growth and induce symptoms of oxidative stress concurrent with the depletion of non-protein thiols and ascorbic acid. For instance, in *Tanacetum parthenium*, gallic and ferulic acids were increased in response to Mn stress [70]. In addition, the effects of arsenic (As) toxicity on the carotenoid and chlorophyll contents were less pronounced in tetraploid rice than in the diploid one, with significant differences in the sucrose and glucose contents between them. Tetraploid rice plants had thicker cell walls than diploids, limiting the entry of heavy metals into the cell membrane and protoplasm, thus serving as a biosorbent for metals and restricting their translocation to other cell tissues [71]. Moreover, autotetraploid rice showed improvements in all physicochemical parameters under cadmium (Cd) stress compared to diploid rice [72]. And by improving the related antioxidant enzyme activities and physiochemical attributes, the application of zinc–oxide nanoparticles (ZnO-NPs) significantly alleviated the Cd toxicity in both lines and eventually significantly improved plant growth and decreased Cd accumulation in different ploidy rice [71]. In addition, the contents of aluminum (Al), Cd, and Cu in the roots of diploid tea varieties were significantly higher than in triploid ones. Furthermore, the contents of Cu in the leaves of diploid tea were significantly higher than that in the roots of triploid tea. And the contents of Al, Cu, and Cd in the soil around the roots of triploid tea were higher than that of diploid tea, showing that the roots of triploid tea had a stronger ability to accumulate metals [73].

#### 3.1.2. Polyploidy and Biotic Stress

Apart from abiotic stresses, in the face of biotic stress, such as microbial disease and pest infestations, crop growth and yield were also significantly impacted in polyploids. However, polyploid plants have been observed to possess a heightened resistance to biotic stress compared to diploids. For instance, tetraploid apple cultivars exhibited a marked reduction in sporulation compared to their diploid counterparts and showed fewer signs of chlorosis and necrosis after being infected with *Venturia inaequalis* [74]. Furthermore, the resistance of sugarcane to smut infection appears to be influenced by several biological processes, such as immune response, protein metabolism, cell wall formation, and polyamine (PAs) synthesis [75]. Polyamines have been implicated in the cross-tolerance of tomato plants by biotic and abiotic stresses, and the accumulation of PAs resulted in reduced H_2_O_2_ levels, which could mediate defense against viral infection. In addition, PAs also play an integral role in the hypersensitive response by contributing to a rapid burst of ROS upon contact with pathogens. This response hindered pathogen establishment and proliferation by generating regulatory signals that stimulate the transcription of defense-related genes [76]. In addition, autotetraploid Arabidopsis plants exhibited constitutively activated defense mechanisms against pathogen infection, and the ABA signaling pathway coincided with reduced post-invasion defense and through regulating stomatal closure in response to microbe-associated molecular patterns (MAMPs), marked by the decreased callose deposition and salicylic acid (SA)-dependent resistance, thereby leading to the enhancement of resistance and improved protection against pathogens [77]. However, many more studies related to biotic stress in polyploids should be conducted to understand the underlying mechanism of polyploid resistance to biotic stress.

In brief, though polyploid plants have specific response pathways for different types of stress, there are common response pathways involved in the regulation of stomatal closure, the AsA–GSH cycle (Halliwell–Asada Pathway) and its metabolism pathway, the active oxygen scavenging pathway, the mitochondrial matrix, and photosynthetic efficiency under different stresses (Figure 2).

### 3.2. Molecular Mechanistically, Polyploidy Relieves Stress

The molecular mechanism underlying the enhanced stress tolerance of polyploid plants can be explained from the following aspects: (1) genome duplication effects; (2) alteration in gene expression; (3) alternative splicing effects; (4) protein interaction modification; and (5) epigenetic modification.

#### 3.2.1. Genome Duplication Affects Polyploidy Stress Tolerance

In polyploid organisms, the expression of duplicated genes often undergoes sub-functionalization. This process leads to a partitioned expression pattern between the duplicates, where one copy is active in certain organs, while the other is expressed in different organs at different times or treatments [78]. Such a mode of expression is particularly evident in response to abiotic stress. For instance, in allopolyploid cotton (*Gossypium hirsutum*), specifically with the alcohol dehydrogenase gene *AdhA*, one copy of *AdhA* was expressed exclusively in the hypocotyls during water-submersion treatment, whereas the other copy was activated only under cold stress [79]. Similarly, with the increase in copy numbers, the heat shock transcription factor (*Hsf*) genes (produced by WGD events) were also coupled with gene sub-functionalization, leading to a broadened temperature range for optimal growth in the Asteraceae family [80]. However, in autotetraploid Arabidopsis, both the formation of tetravalent and successful chromosome pairing were negatively correlated with temperature elevation. This correlation suggests that increased temperature primarily interferes with crossing-over by affecting homolog pairing, indicating that genome duplication does not enhance thermal tolerance in the meiotic recombination of polyploid *A. thaliana* [48].

#### 3.2.2. Alteration in Gene Expression Confers Polyploidy Stress Resistance

Abiotic and biotic stress conditions can significantly affect duplicate gene expression in polyploid organisms, with effects varying by gene, stress, and organ type. Differential gene expression in response to environmental stresses may contribute to the preservation of some duplicated genes in polyploids [81]. For example, in wheat, an increased copy number of *VRN-A1* was linked to enhanced frost tolerance in varieties carrying the *FR-A2-T* allele [82]. In addition, *HSFs* are considered as an integral part of signal transduction pathways against environmental stresses, especially heat stress [83]. In polyploid cotton, redundant duplicates of *HSFs* may be involved in regulating thermotolerance or other abiotic stress tolerance. In addition, in allotetraploid *Gossypium hirsutum*, *GhiHSF14* was upregulated in response to heat and downregulated in salinity, which showed almost similar behaviors under cold and PEG-induced stress. Meanwhile, *GhiHSF21* exhibited downregulation across almost all of these stress conditions [84]. Additionally, the expression of the *TaCER1-1A* gene in hexaploid wheat, which plays a role in cuticular wax alkane biosynthesis, could be induced by drought stress, with its genome containing nine CER1 (complex made of Arabidopsis ECERIFERUM1) homologs, each with characteristic expression patterns [85]. Finally, the enhanced expression of CER1 homologs may improve wheat’s drought resistance.

Furthermore, omega-3 fatty acid (FA) desaturase, *EjFAD8*, in triploid loquats (*Eriobotrya japonica* Lindl.), localized in the chloroplast, showed significant upregulation under low temperatures and potentially contributed to the enhanced cold resistance by increasing lipid unsaturation levels, including sulfoquinovosyl diacylglycerol (SQDG) and chloroplast content [86]. In addition, *BnFLC2* and *BnFLC3* are key *FLC* homologs in *B. napus*, which play a crucial role in responding to cold stress by regulating the flowering time [87]. In addition, lipoxygenase (LOX) catalyzes the oxygenation of polyunsaturated fatty acids into diverse bio-functionally fatty acids (oxylipins), which have important biological functions in response to biotic (pathogens and insects) and abiotic stresses. In *B. napus*, members of *BnaLOX2* exhibited the highest expression in stamens and the lowest in roots and were continuously induced by methyl jasmonate (MeJA) and SA but were strongly repressed by cold, heat, and waterlogging stress in leaves [88]. Under drought stress, a range of genes associated with stress tolerance, such as *RHA2b* (RING-H2 E3 ligase), *CIPK5* (Calcineurin B-like protein (CBLs)-interacting protein kinases), *PIP1* (Plasma membrane intrinsic proteins), *SRP* (stress-related protein), and *EXP1* (Expansin), were more significantly induced in tetraploid *Poncirus trifoliata* than in diploids, regulating stomatal closure, acting as Ca^2+^ sensors, functioning as water channels, encoding stress-related proteins, and promoting cell division and elongation, respectively, among other genes [56]. Moreover, in tetraploid *Lycium ruthenicum*, key enzymes in ABA synthesis, *9-cis-epoxycarotenoid dioxygenase 1* (*NCED1*) and *NCED2*, were significantly upregulated [59]. Another study found that the expression of *NCED* genes was inversely proportional to plant height. Virus-induced gene silencing (VIGS) of *GhNCED1* in cotton led to increased plant height and number of internodes, but it also reduced the plant’s resistance to PEG and salt stress [89]. Additionally, the increased expression of the *ABRE-binding factor 5-like* (*ABF5-like*) gene enhanced the activation of the ABA signaling pathway and its downstream target genes in tetraploid *Lycium ruthenicum* compared to diploids. This upregulation is crucial because ABA significantly induces the expression of osmotic proteins, thereby enhancing plant stress tolerance at the translational level. Consequently, tetraploid plants exhibit superior resistance under severe drought conditions [59]. Furthermore, in Arabidopsis polyploids, the inductions of *WRKY18* and *WRKY40* were correlated with a more rapid expression of *PR1* upon *Pseudomonas syringae* (*Pst*) infection, as well as their direct targets *ISOCHORISMATE SYNTHASE1* (*ICS1*), *ENHANCED DISEASE SUSCEPTIBILITY1* (*EDS1*), and *AVRPPHB SUSCEPTIBLE3* (*PBS3*) involved in SA signaling, compared with their progenitors [9].

Furthermore, wheat plants with VIGS-mediated suppression of *Era1* and *Sal1* exhibited increased relative water content, improved water-use efficiency, reduced gas exchange, and enhanced vigor under drought treatment. This indicates that *Era1* and *Sal1* play a significant role in conferring drought tolerance in wheat [90]. Additionally, the higher expression of transcription factor *IiWRKY34* in tetraploid *Isatis indigotica* confers positive effects on biomass growth rates, lignan biosynthesis, and salt and drought stress tolerance [91]. In addition, arabidopsis polyploids were demonstrated with the enhancement of resistance against the model pathogen *Pseudomonas syringae* pv. tomato DC3000 compared to their diploids, primarily due to upregulated differentially expressed genes (DEGs). These genes included *AMC4*, which played a role in programmed cell death in response to *P. syringae* exposure; *CYP19–1*, which was involved in the production of ROS following *P. syringae* infection; and *VDAC1*, which contributed to the maintenance of ROS homeostasis and stress response [77]. In addition, a newly identified cluster of glutathione S-transferase genes, including *Gh_A09G1508*, *Gh_A09G1509*, and *Gh_A09G1510*, had been demonstrated to confer resistance to Verticillium wilt. Overexpression and suppression of these genes contributed to enhanced resistance in tobacco and significant susceptibility in cotton, respectively [92]. Further investigation revealed that the enzymes encoded by this cluster were crucial for maintaining a balance between the production and scavenging of H_2_O_2_, thereby establishing a new equilibrium state. This function, similar to that of CAT and POD, enhanced the plant’s response to *Verticillium dahliae* [92]. Moreover, *FaRXf1* in octoploid strawberry was identified as conferring resistance against Angular leaf spot, which was caused by *Xanthomonas fragariae* [93]. Additionally, the higher expression of *TaHAG1* in synthetic hexaploid wheat promoted H_2_O_2_ accumulation and the expression of three NADPH oxidase genes, playing a critical role in wheat’s salt tolerance. Consequently, this leads to ROS production and enhanced salt tolerance in wheat [94]. Additionally, the expression of genes involved in redox homeostasis, ABA, and stress response in Arabidopsis tetraploids were primarily regulated under drought stress, leading to the deregulation of numerous genes associated with ABA metabolism and signaling. Consequently, tetraploid seeds exhibited increased sensitivity to ABA during germination, and tetraploid plants experienced delayed flowering [58]. Furthermore, WRKY transcription factors, which are key regulators of ABA signaling [95] and over-represented in autopolyploids of various species [96], are linked to the enrichment of *cis*-elements containing MYB and WRKY binding motifs in the promoters of genes regulated by tetraploids [58]. Furthermore, autotetraploids modulate ABA synthesis by regulating the gene expression of these transcription factors, enhancing drought resistance. Altered expression levels of genes that regulated floral transition, such as *FLC* and *FT*, also contributed to the delayed flowering time observed in tetraploid plants [58].

Under Cu stress conditions, the transcription levels of four ABA-responsive genes (*RD29A*, *RD29B*, *ABI5*, and *RAB18*) were significantly increased in both diploid and tetraploid Arabidopsis, with the expression being notably higher in tetraploids than in diploids after such treatment, indicating that excess Cu induced positive regulation of expression for ABA-response genes in tetraploids [69]. In addition, in Italian ryegrass, the relative expression level of the DEG *LmAUX1* was downregulated under Cd stress, suggesting enhanced self-tolerance through decreased *LmAUX1* expression in this un-sequenced autotetraploid species [97]. In addition, under arsenic (As) stress, the *Os04g0380000* gene was specifically expressed in tetraploid rice, whereas *OsMT1a* was highly expressed in the control of tetraploid rice, potentially leading to stronger As resistance in tetraploid rice. Furthermore, many aquaporins, including *OsPIP1;1* (*Os02g0666200*), *OsPIP1;2* (*Os04g0559700*), and *OsPIP1;3* (*Os02g0823100*), were differentially expressed under As stress between tetraploid and diploid rice. Moreover, the expression levels of tubulin genes such as *OsTUB8* (*Os03g0661300*), *OSTB-50* (*Os02g0167300*), and *OSTB-16* (*Os01g0805900*) were increased in tetraploid rice but decreased in diploid rice under the same conditions. Additionally, the expression of *RSUS1* (*Os06g0194900*) was decreased under As stress, while *RSUS2* (*Os03g0401300*) expression was increased in tetraploid rice but decreased in diploid rice. In total, these differential expression levels of tubulins, aquaporins, transporters, and glutathione metabolism-related genes could be a crucial factor contributing to the better As tolerance of tetraploid rice compared to diploid rice [71]. Moreover, under Cd treatment, the expression levels of *OsSUT1* and *OsSUT2* were also higher in autotetraploid rice. In addition, the *NRAMP* (natural resistance-associated macrophage protein) family, known for its role in heavy metal (HM) transport and featuring membrane integrins, includes *NRAMP1*, *NRAMP2*, *NRAMP3*, *NRAMP4*, *NRAMP5*, and *NRAMP6*, which are involved in metal transport, especially Cd. Notably, the low expression levels of *OsNRAMP1*, *OsNRAMP4*, and *OsNRAMP5* in polyploid rice under Cd stress might be associated with its resistance mechanism. Furthermore, in the family of HM transporters, P1B-type ATPases (HMA) participate in the absorption, transport, and accumulation of metal elements of plants. The expression levels of *OsHMA3*, *OsHMA4*, and *OsHMA5* genes were lower in autotetraploid rice under Cd treatment than in diploid rice, suggesting that polyploid rice could alleviate Cd toxicity by altering the expression levels of sucrose, metal transporters, and resistance-related genes [72]. The gene *ScGH3-1* exhibited downregulation following infection by the smut pathogen in *Saccharum*, yet was upregulated in response to ABA, MeJA, and SA treatments, suggesting that *ScGH3-1* was involved in the plant’s stress response, operating through the ABA, JA, and SA signaling pathways [98].

#### 3.2.3. Alternative Splicing Affects Polyploidy Stress Tolerance

Alternative splicing (AS) is a critical regulatory mechanism in gene expression that enables the production of diverse transcript variants from a single mRNA precursor. In allotetraploid *B. napus* under abiotic stress conditions, the C_T_ subgenome exhibited a notably higher abundance of alternatively spliced isoforms, as well as a greater frequency of various AS events, compared to the A_T_ subgenome [87]. In addition, different quality genes exhibit significant changes in alternative splicing (AS) patterns in response to drought stress (DS), heat stress (HS), and their combination (HD), respectively. Combined drought and heat stress can induce specific AS events compared to individual stressors. Several significant stress-responsive genes have been found to show conserved alternative splicing patterns in response to these stresses across different plant species [99]. Moreover, heat shock proteins (*HSPs*) are subjected to significant AS modulation under DS, HS, and HD conditions in wheat seedlings. In addition, homeologous genes exhibited partitioned AS responses under stress conditions, with more AS events occurring in the B subgenome compared to the A and D subgenome in hexaploid wheat. Thus, this indicated the subfunctionalization or neofunctionalization of homeologous genes in terms of alternative splicing during the polyploidization in wheat history [99]. Moreover, AS might be implicated in the early heat-sensing mechanisms of bread wheat, with the AS response manifested significantly earlier than the HSF response, suggesting that AS regulation operated independently or partially independent of transcriptional regulation, playing a crucial role in the plant’s early response to heat. This finding provided valuable insights for future research on heat sensing and signaling mechanisms [100]. Furthermore, large numbers of genes experienced AS in response to smut infection in sugarcane (*Saccharum* spp.) 200 days after infection (DAI). Genes involved in cell wall modifications, transcription factors, ROS scavenging, and defense signaling were among those that underwent AS during smut infection. In addition, AS in multiexonic genes led to the loss of protein domains, notably in tubulin alpha-5 and the floral homeotic protein (HUA1) [101].

#### 3.2.4. Protein Interaction Increases Polyploidy Stress Tolerance

Apart from gene expression regulation, the physical interaction of different proteins can also affect polyploid stress tolerance. For instance, the EjFAD8 protein boosts the expression of genes associated with low-temperature resistance and participates in the ICE-CBF-COR cold tolerance pathway. Furthermore, EjFAD8 contributes to the stability of photosynthesis in loquats by regulating ROS in chloroplasts, thereby improving stress resistance in triploid loquats [86]. In addition, exposure to ABA or salt triggered the shuttling of NUCLEAR FACTOR Y SUBUNIT C 9 (*PtoNF-YC9*) into the nucleus, where it interacted with the SALT RESPONSIVE MYB TRANSCRIPTION FACTOR (*SRMT*, a *MYB* gene), leading to the rapid expression of *RESPONSIVE TO DESICCATION 26* (*PtoRD26*), which, in turn, directly regulated *SRMT*. This positive feedback loop, *SRMT-P to RD26,* could rapidly amplify salt-stress signaling, thus conferring enhanced salt tolerance in triploid poplar [102]. In addition, the apetala2/ethylene response factor transcription factor, INDETERMINATE SPIKELET1 (IDS1), was found to engage in the physical interaction with the transcriptional corepressor topless-related 1 and the histone deacetylase HDA1. This interaction plays a key role in suppressing the expression of *LEA1* (*LATE EMBRYOGENESIS ABUNDANT PROTEIN1*) and *SOS1* (*SALT OVERLY SENSITIVE1*). Additionally, an in-depth analysis of the histone H3 acetylation status and RNA polymerase II occupation at the *LEA1* and *SOS1* promoters has elucidated the underlying mechanisms of IDS1’s role in transcriptional repression [103]. Moreover, regular loading of chromosome axis proteins ASY1 and ASY4 was crucial in maintaining meiotic stability in tetraploid Arabidopsis, which was highly sensitive to temperature changes [48]. In addition, differentially expressed proteins of stress responses and abiotic or biotic stimulus were significantly enriched between *A. thaliana* diploids (At2) and autotetraploids (At4), with 24 of 36 proteins exhibiting higher accumulation levels in At4 compared to At2 [104]. Moreover, the sugar transporter, identified as a differentially abundant protein (DAP), was upregulated under salt stress in the leaves of both diploid and autotetraploid Paulownia fortune, with a total of 152 autotetraploid-specific DAPs being identified. The upregulation of the sugar transporter might enhance the accumulation of soluble sugars, potentially altering osmotic pressure. Additionally, the increased expression of ATP synthase delta in ATP production enhanced ion transport activity and potentially contributed to the establishment of a proton gradient in response to salt stress [105].

#### 3.2.5. Epigenetic Modification Affects Polyploid-Induced Stress Tolerance

Polyploidization leads to genome rearrangement and induces a large number of epigenetic modifications, which result in different DNA methylation levels in polyploids compared to their diploid relatives, thus providing a clear distinction in their epigenetic landscapes [106]. Thus, DNA methylation patterns change in response to abiotic stresses, and the cytosine methylation patterns can affect gene expression. Consequently, it is plausible that changes in cytosine methylation patterns between paired duplicate genes may alter the expression levels of one or both genes under abiotic stress conditions [81]. Different ploidies often occupy distinct niches and exhibit varying methylation levels. For instance, tetraploid (4x) plants of *Ranunculus kuepferi* found at the highest elevations were demonstrated to have more dynamic methylation changes when compared to the warm-adapted diploid progenitors [107]. Furthermore, tetraploid rice exhibited enhanced salinity tolerance compared to diploid rice, attributed to lower sodium uptake and the epigenetic regulation of jasmonic acid (JA)–related genes. Tetraploidy triggered DNA hypomethylation, enhancing the potential for genomic loci to interact with stress-responsive genes, including those involved in jasmonate biosynthesis and signaling pathways for rapid and robust responses under stress conditions. Following salt stress, the upregulated expression of salt-responsive genes led to hypermethylation, thereby suppressing adjacent transposable elements (TEs). This intricate feedback mechanism, involving polyploidy-induced DNA hypomethylation for swift and potent stress response, coupled with stress-induced hypermethylation to repress TEs and/or TE-associated genes, may confer evolutionary advantages, fostering enhanced adaptation in polyploid plants and crops [108]. In addition, under cold stress, autotetraploids of *Poncirus trifoliata* exhibited a higher accumulation of unsaturated fatty acids (UFAs) and JA than their diploid counterparts. FAD, a key enzyme in UFA biosynthesis, can adjust membrane fluidity to enhance cold stress tolerance. Notably, in autotetraploids, *PtrFAD7* showed reduced CHH methylation in its downstream region, correlating with increased expression levels under cold stress. Similarly, the decrease in CHH methylation downstream of two crucial JA synthesis genes, PtrLOX3 and PtrAOC2, also led to their upregulation in tetraploids. This pattern suggested that tetraploids, due to more extensive DNA demethylation, synthesized UFAs more effectively and relayed JA signaling more efficiently, mounting a more intense response to cold stress in tetraploids than in diploids [109]. In addition, exposure to stressful environments induced epigenetic alterations that might persist even after the stress has subsided, thus establishing a ‘stress memory’. This memory could remain stable throughout an organism’s lifespan and might even be transmitted across generations, a phenomenon particularly notable in plants [110,111].

Furthermore, the histone modification landscape could also be affected in polyploids under stress. For example, studies showed that there is a genome-wide enrichment of H3K27ac in gene regions following WGD [43]. In addition, histone acetyltransferase TaHAG1 contributed to plant salt tolerance by modulating ROS production and signal specificity. Furthermore, TaHAG1 directly targeted a subset of genes responsible for hydrogen peroxide production, and its enrichment led to the increased H3 acetylation and transcriptional upregulation of these loci under salt stress. Additionally, it is found that the salinity-induced TaHAG1-mediated ROS production pathway played a role in the different salt tolerance among wheat accessions with varying ploidy levels [94].

Moreover, one study demonstrated that miRNAs might mediate plant responses to abiotic stresses by regulating the expression of targeted stress-related genes. Numerous candidate miRNA target genes are associated with the ICE1–CBF pathway, a key regulator of freezing tolerance in cold-hardy plants. These targets included DEAD-box ATP-dependent RNA helicase 12, CBF, and dehydrin. Notably, the miRNA candidate apMir_16808, which is responsive to cold conditions, can specifically target cold-responsive genes such as dehydrins, playing a pivotal role in the freezing tolerance mechanism of hexaploid wheat [112].

In addition to these, other type of epigenetic modifications can also affect polyploid stress tolerance. For example, in *Citrus wilsonii*, tetraploids might activate protein kinase activity in response to drought stress, leading to phosphorylation modifications, such as dephosphorylation, autophosphorylation, and either negative or positive regulation through phosphorylation. Furthermore, under drought conditions, 93 proteins in tetraploids underwent phosphorylation modifications, which were primarily involved in signal transduction, post-translational modifications (PTMs), and chaperones [53]. Moreover, under abiotic stress, the expression level of the *MdHAL3* gene in autotetraploid apples was significantly upregulated, exerting 4′-phosphopantothenoylcysteine decarboxylase (PPCDC) activity. Furthermore, both dephosphorylated CoA synthase (PPAT) and *HAL3*, catalytic enzymes were involved in coenzyme A (CoA) synthesis. Thus, the upregulation of *HAL3* increased the level of CoA in plants, thereby enhancing the salt tolerance of autotetraploid apples [113].

After polyploidization, the shared patterns of morphological and physiological alterations across diverse cellular, tissue, and organismal contexts in polyploid plants have yet to be gradually comprehended [114]. However, extensive investigation is still warranted to elucidate the exact molecular and genetic mechanisms underpinning the morphological and physiological transformations ensuing species polyploidization (Figure 3).

In summary, the integral part of signal transduction pathways against environmental stresses (HSFs), the biosynthesis of coenzyme A (CoA) from pantothenic acid (vitamin B5), ABA synthesis, and the ABA signaling pathway were the common regulatory pathways under different stresses. Subsequently, genetic modification (genome doubling, alteration in gene expression, and alternative splicing), protein interaction modification, and epigenetic modification all affect polyploid resistance to different stresses in different aspects. Therefore, with the development of biological technology, RNA-seq, ChIP-seq, ATAC-seq, Hi-C, Hi-ChIP data, and other technology can be integrated into future analysis for the underlying mechanism of polyploid stress tolerance (Figure 3).

## 4. Enhanced Stress Resistance Is Probably Independent of Polyploidy

In most instances, polyploidization enhances plant resistance; however, there are cases in which plant resistance is not directly related to ploidy. For example, salt tolerance of heterologous polyploid materials derived from two diploid cotton genomes showed no significant differences, suggesting that the presence of heterologous polyploids alone did not necessarily confer increased salt tolerance [115]. Similarly, under conditions characterized by low nitrogen and phosphorus concentrations, haploid prokaryotes were found to have advantages over polyploid species [116]. In addition, in the eastern Mediterranean, drought did not affect the proportion of allopolyploid and diploid individuals within populations. Instead, the phenology of desert plants, particularly in short-stem grass, was primarily regulated by phenotypic adjustments rather than ploidy levels, indicating an association with drought adaptations [54]. In addition, genome doubling in the first-generation synthetic self-pollinated *Fragaria* lines did not substantially improve the average heat tolerance [117]. Furthermore, polyploidization in *Solidago canadensis* led to a decrease in freezing tolerance. This reduction was attributed to promoter methylation repressing the expression of the multi-copy *ScICE1* genes, resulting in reduced freezing tolerance in polyploid *S. canadensis* compared to its diploid counterparts [106].

Generally speaking, polyploidy exhibits a slower rate of speciation and a heightened rate of extinction, thus resulting in lower diversification rates among polyploids when compared to their diploid counterparts [5,118].

## 5. Conclusions and Future Prospects

Most polyploid plants originated from abiotic stress and could have abiotic and biotic stress resistance. However, the underlying mechanisms of the cellular processes and physiological and biochemical characteristics in polyploid plants in response to environmental stressors have been a mystery for centuries. A review can not answer all questions; more research is needed to further understand these mechanisms. Therefore, we speculated that genome structural alteration, gene expression regulation, post-transcriptional, post-translational modification, epigenetics, and maybe other factors all affect the actual gene expression and function protein in response to different stressors in polyploids after polyploidization. With the advancement of different biotechnologies and other related technologies, the general gene/genome structural variation and gene expression regulation, as well as other deeper mechanisms of alteration splicing, protein interaction, DNA methylation, and other post-transcriptional, post-translational modification, and epigenetics, will be investigated thoroughly for the stress resistance/tolerances in polyploid plants. Furthermore, we found that studies focusing on biotic stress in polyploids are less common than those focusing on abiotic stress, indicating that more research should be conducted in the future.

## Figures and Tables

**Figure 1 ijms-25-01957-f001:**
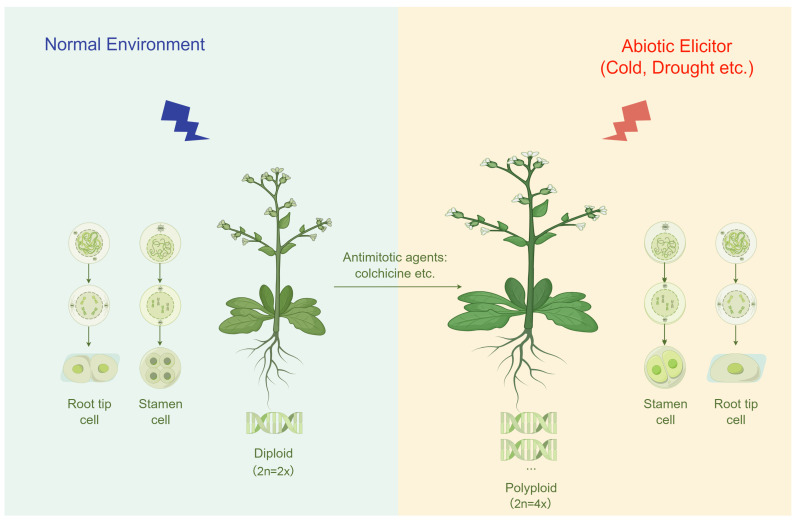
A working model of polyploids produced by abiotic stresses and chemical reagents. By Figdraw.

**Figure 2 ijms-25-01957-f002:**
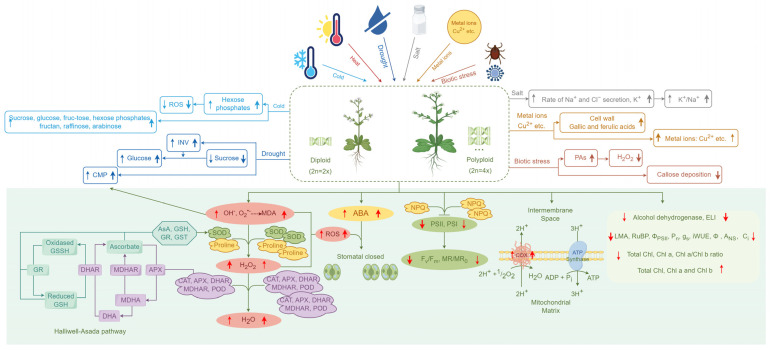
A working model for cellular, physiological, and biochemical mechanisms of polyploid tolerance under stress. Arrows indicate the increased or decreased levels of biological factors, with the thickness of the arrows denoting the amount of the increase or decrease, and the alteration of the diploid is represented on the left and polyploid on the right. The pathways with light green background at the bottom of the figure illustrate the common regulatory pathways under different stresses, while the unique regulatory mechanisms specific to different stress conditions are depicted with the other colors. Activation is denoted by arrows, and inhibition by T-shaped arrows, consistently applied throughout the figures. MDA: malondialdehyde, SOD: superoxide dismutase, ROS: reactive oxygen species, ABA: abscisic acid, GSH, GSSH: glutathione, GR: glutathione reductase, DHAR: dehydroascorbate, MDHAR: monodehydroascorbate, DHA: dehydroascorbate, CAT: catalase, APX: ascorbate peroxidase, AsA: ascorbic acid, GST: glutathione S-transferase, CMP: cell membrane permeability, POD: peroxidase, NPQ: non-photochemical quenching, PSII: photosystem II, PSI: photosystem I, F_v_/F_m_: chlorophyll fluorescence, MR/MR_0_: the maximal photochemical capacity of PSI, COX: Ctochrome c oxidase, chlorophyll: Chl, ELI: electrolyte leakage index, LMA: leaf mass per area, RuBP: ribulose biophosphate, Φ_PSII_: photosystem II, P_n_: net photosynthesis, *g*_s_: stomatal conductance, iWUE: intrinsic water-use efficiency, *Φ*: net CO_2_ fixation, *A*_NS_: the rate of light-saturated net photosynthesis, C_i_: intercellular CO_2_ concentration, DM: dry matter, INV: invertase, PAs: polyamines. By Figdraw.

**Figure 3 ijms-25-01957-f003:**
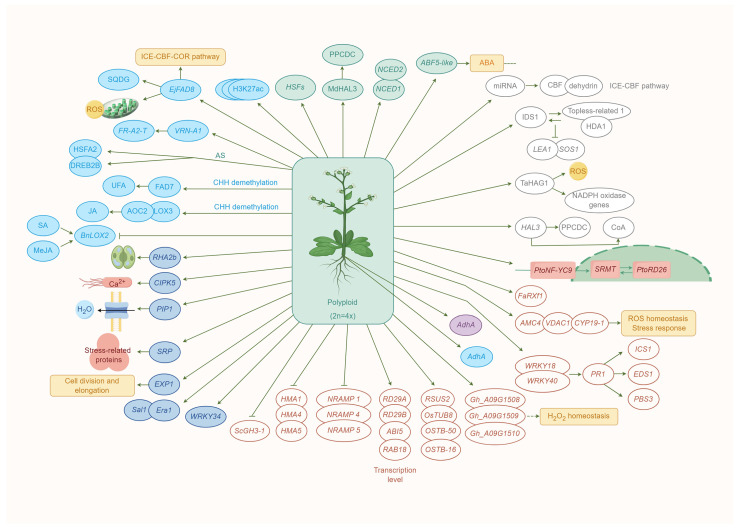
A working model for the molecular mechanism of polyploid tolerance under stress. The common regulatory pathways under stress are illustrated with green fonts, while the unique regulatory pathways specific to different stress conditions are depicted with other colors (the same color usage as that in Figure 2). SA: salicylic acid, *HSFs*: heat shock transcription factors, *FAD*: Omega-3 fatty acid (FA) desaturase, SQDG: sulfoquinovosyl diacylglycerol, LOX: lipoxygenase, MeJA: methyl jasmonate, *RHA2b*: RING-H2 E3 ligase, *CIPK5*: calcineurin B-like protein (CBLs)-interacting protein kinases, *PIP1*: plasma membrane intrinsic proteins, *SRP*: stress-related protein, *EXP*: Expansin, *NCED1*: *9-cis-epoxycarotenoid dioxygenase 1*, *ABF5-like*: *ABRE-binding factor 5-like*, *ICS1*: *ISOCHORISMATE SYNTHASE1*, *EDS1*: *ENHANCED DISEASE SUSCEPTIBILITY1*, *PBS3*: *AVRPPHB SUSCEPTIBLE3*, *OsTUB8* (*Os03g0661300*), *OSTB-50* (*Os02g0167300*), *OSTB-16* (*Os01g0805900*), *RSUS2* (*Os03g0401300*), *PtoNF-YC9*: *NUCLEAR FACTOR Y SUBUNIT C 9*, *SRMT*: *SALT RESPONSIVE MYB TRANSCRIPTION FACTOR*, *PtoRD26*: *RESPONSIVE TO DESICCATION 26*, IDS1: INDETERMINATE SPIKELET1, HAD: histone deacetylase, *LEA1*: *LATE EMBRYOGENESIS ABUNDANT PROTEIN1*, *SOS1*: *SALT OVERLY SENSITIVE1*, *NRAMP*: natural resistance-associated macrophage protein, *ScGH3*: sugarcane *GH3*,HMA: P1B-type ATPases, UFAs: unsaturated fatty acids, JA: jasmonic acid, AOC2: allene oxide cyclase, TaHAG1: histone acetyltransferase TaHAG1, ICE: inducer of CBF expression1, CBF: C-repeat (CRT)-binding factors, COR: cold responsive, PPCDC: 4′-phosphopantothenoylcysteine decarboxylase, CoA: coenzyme A, *HAL3*: halotolerance 3. By Figdraw.

## Data Availability

All data supporting the findings of this study are available in the paper.

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
