# Peer review of "Polyploidization: A Biological Force That Enhances Stress Resistance"

_ijms, 2024, doi:10.3390/ijms25041957_

Round 1
Reviewer 1 Report
Comments and Suggestions for Authors
Dear authors!
I am not a specialist in polyploidy, but I understand something about molecular biology of plants and plant stress tolerance, so I am very interested in the subject of this review. For a better understanding of the issue I read several other reviews of the last years on this topic. As a result, I came to the conclusion that the review is interesting and I quite admit that it will be in demand after publication in the journal.
The issue of polyploidy is very important from the standpoint of evolution, biotechnology, breeding, and plant physiology, especially in connection with mechanisms of plant stress tolerance. All the reviews I have read, with varying degrees of detail, address more or less similar issues of polyploidy. The peculiarity of this review lies in the selection and analysis of a large amount of data, mainly from recent years, proving the great resistance of polyploid plants to stressors of different nature. The main, but by no means all, abiotic stressors are considered and the resistance of polyploids to biotic stressors is analysed much more briefly (incidentally, this issue is presented in more detail in another review by Yves Van de Peer et al. PLANT CELL 2021: 33: 11-26). One cannot but agree with the authors that the review can serve as a reference for many subsequent studies. For the abiotic stressors reviewed, many data are provided that characterise the plant response under different types of stresses, such as enzyme activities, expression of stress-inducible genes, photosynthetic pigment content, photosystem status, stress phytohormone biosynthesis enzymes (ABA, SA and JA) and so on.
The authors of this review, as well as several other reviews, conclude that the greater stress tolerance of polyploid plants can be explained by: 1. genome duplication effects; 2. changes in gene expression; 3. alternative splicing effects; 4. modification of protein interactions and 5. epigenetic modification. For each of these positions, the currently available literature is cited. The first two positions have fairly strong support, while the last three are more problematic due to the lack of data available. In any case, this is a work in progress and significant progress is possible in the future.
I have virtually no comments. The authors planned to write such a review and it is not necessary, in my opinion, to demand from them to expand the questions of evolutionary origin of polyploids or resistance to biotic stress. I would like to draw attention to very informative Figs. 2 and 3. If this review reaches polyploidy professionals, I hope they will provide useful specific advice.
From very minor comments. In some cases, the authors refer generally to three abiotic stressors (temperatures, drought, and salinity). The sentence is constructed as if there are no other abiotic stressors.
The following expression seems to fail. "Stress resistance is probably independent of polyploidy" (line 665). There is no doubt that the basic mechanism of stress resistance is independent of polyploidy, if only because many diploid plants are resistant to many stressors. Polyploidy complicates, modifies and alters the stressor response mechanism in such a way that plants become more resistant.
I recommend that the review be accepted virtually unchanged. I ask the authors to address my second point.
Author Response
Thanks a lot for your comments.
Firstly, due to the fast advancement of biotechnology, there was no doubt that the research on the last three positions (alternative splicing effects, modification of protein interactions, and epigenetic modification) related to greater stress tolerance of polyploid plants would be in progress and with greater significance in the future. However, to make this review more current, we have consulted references for more data about the last three positions and have added some new insights on these aspects in the revised manuscript.
Secondly, the reason why we expand a little about the questions of the evolutionary origin of polyploids is that we want to draw a simple conclusion that almost all polyploids originated from abiotic (either environmental or artificial) stress. Then, the reason why we discuss the resistance of polyploids to biotic stress is that we want to know whether there are some common or different pathways in response to biotic and abiotic stressors, to shed some light on really professional researchers in this field. Unfortunately, data from biotic stress are few, so we could not reach a clear conclusion. Therefore, we show some simple responses to biotic stress in polyploids. The detailed thought of this review starts from the evolutionary origin of polyploids, the phenotype of polyploid plants, then to the cellular, physiology, and biochemical characteristics, and last but not end, to decipher the underlying molecular mechanisms of biotic and abiotic stress resistance or tolerance after polyploidization. Finally, we conclude that polyploidization is a biological force that enhances stress resistance as shown in Figs. 2 and 3. In summary, with the advancement of biotechnology and other related technology, further deeper decipherment of gene/genome structural variation and gene expression regulation, together with the post-transcriptional, post-translational, and epigenetics mechanisms underlying the fascinating stress resistance/tolerance after polyploidization will be much more helpful for crop improvement in the future.
Thirdly, to our knowledge, more researchers are concentrated on stressors like temperatures, drought, and salinity, and there is more data about these three abiotic stressors. We also described some other abiotic stressors, like heavy metal ions stress. However, there are few related studies about polyploids under other variety conditions, as well as the underlying molecular mechanism. To make this manuscript more current, we have consulted more references and have refreshed them in the revised manuscript.
Finally, we also appreciate your suggestion regarding stress resistance and polyploidy. Perhaps the statement “Enhanced stress resistance is probably independent of polyploidy” is closer to the current understanding, and we have accordingly updated the manuscript (line 665).
We appreciate your suggestions on stress resistance and polyploidization. Thanks again for your kind contribution to this manuscript.

Reviewer 2 Report
Comments and Suggestions for Authors
Polyploidization: A biological force that enhances stress resistance
Thus, this review summarizes the mechanism of polyploid formation, polyploid biochemical tolerance mechanism to abiotic and biotic stressors, and molecular regulatory networks that confer polyploidy stress tolerance, which can shed light on the theoretical foundation for future research. This review article is essential to complete the underlying stress tolerance mechanisms of polyploids. The underlying mechanism of the cellular processes, physiological and biochemical characteristics, and molecular regulatory mechanisms like genome duplication, gene expression, alternative splicing, protein interaction, and epigenetic modification were discussed here. However, the underlying mechanism of stress tolerance in polyploidy remains to be decoded deeper, necessitating continuous research. Additionally, the authors found that studies focusing on biotic stress in polyploids are less common than those on abiotic stress, indicating that more future research should be conducted.
Actually, this is a piece of work that demonstrates dedication and good form. It is very easy to read and quite clear. The MS is interesting and well written. Few comments and suggestions are as follows:
- General comment: From your expertise is there any relation between polyploid and salt responsive proteins that are potential determinants for plant salt adaptation? Is there any relation between changes in proteome composition and abundance, associated with salt tolerance and polyploid?
- Line 258. Concerning salt stress. Soil salinity is an important abiotic stress that limiting plant growth and development, negatively influencing morphology, photosynthetic efficiency, chloroplast structure, and mitochondria function. This information need reference. I suggest the following one:
Mansour, M.M.F., Hassan, F.A.S., 2022. How salt stress‑responsive proteins regulate plant adaptation to saline conditions? Plant Molecular Biology 108:175–224 https://doi.org/10.1007/s11103-021-01232-x
- Line 283. Elevated accumulations of H2O2 and malondialdehyde (MDA) were also observed under salt stress. The more H2O2 accumulation often followed with the significant increase of essential antioxidative enzymes and non-enzymatic antioxidants. This information need reference. I think the same suggested reference can fill the gap.
- Line 368. However, much more studies related to biotic stress in polyploids should be conducted for the underlying mechanism of polyploid resistance to biotic stress (Figure 2). I think there is no relation between the text and Fig. 2. The authors mentioned it in correct place Line 293.
- Line 685. Conclusions. I think this part is a little bit long. Besides, it contains some references as well as Figs. and this part should not contain any reference or Figs. Please focus only on the message that you want to deliver.
Lastly, I would like to mention that this is a very good piece of work that complies with the standards of a MDPI scientific journal.
Author Response
Thanks for your comments. There is no doubt that gene expression contributes to protein composition and abundance. However, due to the advancement of biotechnology and life science, more and more studies have shown that protein composition or abundance had little positive relation to the transcript in common plants, even polyploids under environmental stresses. So, we speculated that there is some relation between changes in proteome composition and abundance associated with salt tolerance and polyploids, but more future studies are needed. To make the speculation more solid, we have consulted more references for evidence and have refreshed the revised manuscript. Besides, we have added the new citations (Mansour et al., 2022, and others) in the revised manuscript. Figure citations in Line 368 and Line 293 were also refreshed.
Your comments do help the modification of this manuscript, drawing a more clear conclusion. We have thoroughly refreshed our conclusion and removed the unnecessary citations in the revised manuscript.
Thanks again for your contribution to this manuscript.

Reviewer 3 Report
Comments and Suggestions for Authors
The paper discusses the cellular, physiological, biochemical, and molecular mechanisms of polyploid tolerance under different stress conditions. It elaborates on how polyploidization, which involves genome doubling, leads to changes in gene expression, alternative splicing, protein interactions, and epigenetic modifications, thereby enhancing plant stress resistance. There are some suggestions:
The paper predominantly focuses on abiotic stress. Including more information on biotic stress in polyploids could provide a more balanced view and address a gap mentioned in the paper.
Include more case studies to support the findings. This could involve detailed analysis of specific examples where polyploidy has played a notable role in stress tolerance.
A comparative study between polyploid and non-polyploid species under similar stress conditions might provide insightful data on the exact benefits of polyploidy.
Delve deeper into the molecular mechanisms. For instance, how exactly does genome duplication affect gene expression at the molecular level in the context of stress response?
Update the literature review to include the most recent research, ensuring that the paper reflects the current state of knowledge in the field.
Comments on the Quality of English Language
The English used in the paper is well-suited for a high-level academic publication in the sciences. The technical proficiency in language reflects the complexity and depth of the subject matter.
Author Response
Thank you for your contribution to this manuscript.
Firstly, there was no doubt that including more information on biotic stress in polyploids could provide a more balanced view. However, almost all the studies on biotic stress polyploids were harvested in this review. Abiotic stress can be controlled well (more studies with a deeper understanding view), while biotic stress is particularly uncontrollable (few studies with a more superficial perspective). We have consulted again for biotic stressor-related references and refreshed them in the revised manuscript.
Secondly, it was actual that a comparative study between polyploid and non-polyploid species under similar stress conditions might provide insightful data on the exact benefits of polyploidy. Most of the studies cited in this review were conducted under the same conditions with plants of different ploidy levels.
Thirdly, your comments about how genome duplication affects gene expression at the molecular level are meaningful to us. For a deeper description of how genome duplication affects gene expression at the molecular level in the context of stress response, please refer to reference:
Rehman A, Atif RM, Azhar MT et al. Genome wide identification, classification and functional characterization of heat shock transcription factors in cultivated and ancestral cottons (Gossypium spp.). Int J Biol Macromol. 2021 Jul 1;182:1507-1527. doi: 10.1016/j.ijbiomac.2021.05.016.
Kong X, Zhang Y, Wang Z et al. Two-step model of paleohexaploidy, ancestral genome reshuffling and plasticity of heat shock response in Asteraceae. Hortic Res. 2023 Apr 19;10(6):uhad073. doi: 10.1093/hr/uhad073.
It was true that polyploidization or genome duplication alters the gene/genome structure of plant gene/genome composition and contributes to gene function differentiation (neo-functionalization, sub-functionalization, and non-functionality). You can find the description of this in the introduction part. A review can not solve all questions; more research is needed on the way to the possible truth. However, we speculated that genome structural alteration, gene expression regulation, post-transcriptional, post-translational modification, epigenetics, and maybe other factors all affect the actual gene expression and function protein in response to different stressors in polyploid plants after polyploidization. A more comprehensive whole genome resequencing and transcriptome analysis with a proper polyploid population might solve some concerns on this issue.
Finally, to our knowledge, we have collected all current related papers. As you have referred, we have consulted more related research and refreshed them in the revised manuscript. Thanks again for your comments.

Reviewer 4 Report
Comments and Suggestions for Authors
My only suggestion would be to make it more clear which conclusions were reached by the original authors versus those reached by the writers of this review. I often could not tell which was the case. I did not always find the proposed explanations for responses convincing, and would like to know whether the explanations were those of the original authors, or those of these reviewers.
Author Response
Thank you for your contribution to this manuscript. To our knowledge, most studies by the original authors only concentrated on one or two specific aspects of polyploidization-induced stress tolerance. In this review, we came up with the conclusion: (1) Most polyploid plants originated from abiotic stress and could have abiotic and biotic stress resistance; (2) The generally cellular, physiological, and biochemical characteristics and the possibly underlying molecular mechanisms were well summarized according to the current studies; (3) With the advancement of different biotechnology and related other technologies, not only the general concerned gene/genome structural variation and gene expression regulation but also other deeper mechanisms of alteration splicing, protein interaction, DNA methylation, and other post-transcriptional, post-translational, and epigenetic modifications will be investigated thoroughly for the stress tolerance in polyploid plants. We have refined the conclusion part according to our unique opinion and removed the needless cited references in the revised manuscript.
Thanks again for your kind comments.
